# Hapten-mediated recruitment of polyclonal antibodies to tumors engenders antitumor immunity

Brett Schrand[1], Emily Clark[1], Agata Levay[1], Ailem Rabasa Capote[1], Olivier Martinez[1], Randall Brenneman[1], Iris Castro[1] & Eli Gilboa[1]

Uptake of tumor antigens by tumor-infiltrating dendritic cells is limiting step in the induction of tumor immunity, which can be mediated through Fc receptor (FcR) triggering by antibody-coated tumor cells. Here we describe an approach to potentiate tumor immunity whereby hapten-specific polyclonal antibodies are recruited to tumors by coating tumor cells with the hapten. Vaccination of mice against dinitrophenol (DNP) followed by systemic administration of DNP targeted to tumors by conjugation to a VEGF or osteopontin aptamer elicits potent FcR dependent, T cell mediated, antitumor immunity. Recruitment of αGal-specific antibodies, the most abundant naturally occurring antibodies in human serum, inhibits tumor growth in mice treated with a VEGF aptamer–αGal hapten conjugate, and recruits antibodies from human serum to human tumor biopsies of distinct origin. Thus, treatment with αGal hapten conjugated to broad-spectrum tumor targeting ligands could enhance the susceptibility of a broad range of tumors to immune elimination.

---

[1] Department of Microbiology & Immunology, Dodson Interdisciplinary Immunotherapy Institute, Sylvester Comprehensive Cancer Center, Miller School of Medicine, University of Miami, Miami, FL 33136, USA. Correspondence and requests for materials should be addressed to E.G. (email: egilboa@med.miami.edu)

The ability of dendritic cells (DC) to take up tumor antigens is a pivotal step in the induction of antitumor immunity, underscored by studies showing that antibody blockade of CD47, an inhibitory "don't-eat-me" receptor upregulated on many tumor cells, potentiates the induction of protective antitumor T cell immunity in mice[1,2]. That tumor cells have elaborated mechanisms to limit their phagocytic uptake does not mean that the phagocytic process itself is optimal. An efficient mechanism of tumor cell phagocytosis is mediated by Fc receptor (FcR) uptake of antibody-coated tumor cells. Dictated by the nature of the FcR and Fc portion of the antibody, engagement of FcRs by cell-associated antigen–antibody complexes can trigger complement, cell-dependent killing (ADCC), or phagocytosis (ADCP)[3–5]. FcR-mediated functions have an important role in the mechanism of action of therapeutic monoclonal antibodies targeting CD20 (Rituximab), Her2 (Trastuzumab), and EGFR (Cetuximab)[5], and the marked effects of antibodies targeting CTLA-4 or PD-L1 may be mediated via Fc–FcR depletion of intratumoral Treg[6–8] and F4/80+ myeloid cells[9], respectively. Engagement of the FcRs on DC not only triggers the uptake of the antibody-coated cells but also induces their maturation leading to the stimulation of T cell responses against the captured antigens and antitumor immunity, known as the "vaccinal" effect[5,10–14]. A vaccinal effect might be also contributing to the clinical efficacy of antitumor antibodies like anti-MUC1, anti-HER/neu (Trastuzumab), or anti-CD20 (Rituximab)[15–17].

Considerable efforts in academia and industry are devoted to improving the effector function of therapeutic antibodies by optimizing their Fc portion[5,18]. A limitation of using monoclonal antibodies in cancer therapy stems from emerging evidence that point to the importance of combinatorial contribution of FcRs with complementary mode of action. For example, effective T cell responses against antibody-coated tumor cells were shown to be dependent on both ADCC by FcγRIIIA expressing macrophages and subsequent uptake by FcγRIIA expressing DC[14]. Other studies have shown that IgE antibodies, both endogenous and exogenously administered that coat tumor cells, potentiated antitumor immunity via signaling through the high affinity IgεRI receptor expressed on eosinophils[19,20]. Thus, coating tumor cells with antibodies with complementary Fc-encoded effector functions could improve control of tumor growth. Yet, the use of two or more monoclonal antibodies with complementary mode of action will be increasingly challenging in a clinical setting.

Low and colleagues have shown that recruiting anti-fluorescein (FITC) polyclonal antibody to folate receptor expressing tumor cells in mice by coating tumor cells in situ with systemically administered folate-targeted FITC inhibited tumor growth in a T cell-dependent manner[21]. Spiegel and colleagues have shown that DNP conjugated to a PSMA-binding small molecule recruits DNP-specific antibodies to PSMA expressing LnCAP tumor cells, mediates their lysis in an ADCC dependent manner and inhibits LnCAP tumor growth in DNP vaccinated humanized mice[22,23]. Investigators at Altermune Inc. have conjugated an alpha-gal trisaccharide (agal), a hapten that binds to naturally occurring antibodies in humans, to a DNA aptamer that binds to the surface-anchored M protein of group A Streptococcus (GAS), and showed that binding of the GAS-agal conjugate to Streptococcus bacteria recruits anti-agal antibodies from human sera to the bacteria that are taken up and killed by human phagocytes[24]. In a recent study, Carmi et al. have shown that the efficient rejection of minor histocompatibility (miMHC) mismatched tumors, essentially syngeneic tumors expressing foreign antigens on the cell surface, was mediated by pre-existing polyclonal antibodies present in the mouse. The antibodies

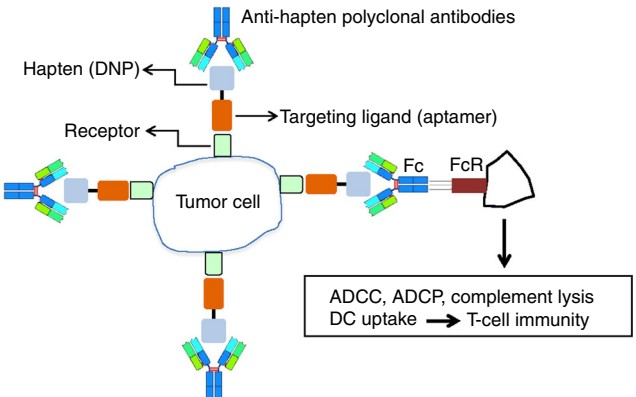

**Fig. 1** Coating tumor cells in situ with endogenous polyclonal antibodies. **a** Hapten-specific polyclonal antibody response is elicited in mice by vaccination, for example, by vaccination against DNP. The vaccinated mice are then administered with the said hapten conjugated to a tumor targeting ligand, for example, an oligonucleotide aptamer, that recognizes a product that is preferentially expressed on the tumor cells (receptor). Consequently, the hapten will accumulate in the tumor and attract the preformed polyclonal antibodies, essentially "coating" the tumor cells with antibodies. The antibody-coated tumor cells will in turn attract Fc receptor expressing phagocytic and dendritic cells, resulting in ADCC-, ADCP-, complement-dependent lysis or uptake and processing for antigen presentation, respectively

recognized the miMHC mismatched tumors on the implanted tumor cells, which led to their enhanced FcR-mediated uptake by DC and stimulation of a strong antitumor T cell response[25]. It is tempting to speculate that the polyclonal nature of the miMHC-specific antibodies capable of engaging multiple FcRs of complementary mode of action, and thereby providing the synergy discussed above, contributed to the remarkable antitumor effects seen in this study. Here we describe a simple and broadly applicable approach to coat tumor cells in situ with endogenous polyclonal antibodies that elicits a strong T cell-dependent vaccinal effect leading to inhibition of tumor growth.

## Results

**Hapten-mediated recruitment of polyclonal antibodies to tumors.** The approach to coat tumor cells with endogenous polyclonal antibodies is depicted in Fig. 1. First, the patient (or experimental mouse) is vaccinated against a harmless or beneficial antigen (e.g., influenza or tetanus toxoid antigens), or a simple hapten-like dinitrophenol (DNP), to generate a polyclonal antibody response. In parallel, the hapten or antigen is conjugated to a ligand that targets the hapten/antigen to the tumor. Ligands can be antibodies, peptides, or oligonucleotide aptamers as used in this study. Next, the ligand–hapten/antigen conjugate is injected systemically into the cancer patient or the tumor-bearing mouse that will target the hapten to the tumor which in turn will attract and opsonize the tumor cells with the preformed hapten/antigen-specific polyclonal antibodies, as seen by Carmi et al.[25] using the miMHC mismatch tumor model. Ultimately, as discussed below, one can exploit pre-existing naturally occurring polyclonal antibodies and dispense with the vaccination step altogether.

**Tumor-targeted DNP administration mice induces antitumor immunity.** We first tested whether DNP hapten-specific antibodies can be recruited to B16.F10 melanoma tumor cells engineered to express a mutant human PSMA that has a deletion in

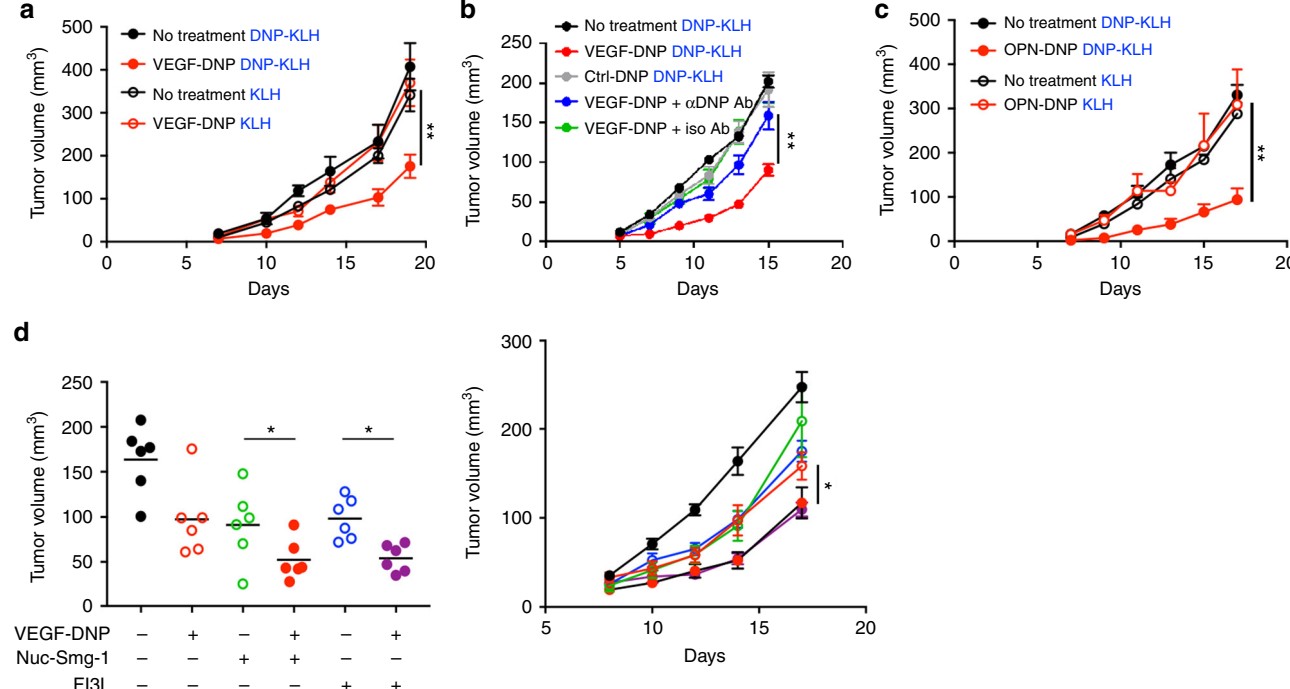

**Fig. 2** Inhibition of tumor growth in DNP immune mice by targeting DNP to palpable subcutaneous 4T1 tumors. **a** Balb/c mice vaccinated against DNP (DNP-KLH) or mock vaccinated (KLH) were injected subcutaneously with 4T1 tumor cells. 7–8 days later when tumors became palpable 500 pmoles of VEGF–DNP was administered via i. p. injection three times 3 days apart and tumor growth was monitored. Statistical analysis, VEGF–DNP in DNP-KLH versus KLH, **$p = 0.0081$, Student's $t$-test. (7 mice per group) ($n = 2$). **b** DNP-vaccinated mice were also treated with control aptamer conjugated to DNP (Ctrl-DNP) or non-vaccinated mice were injected with a mixture of VEGF–DNP and monoclonal anti-DNP or isotype antibody at a ratio of 1:2. (6 mice per group) Statistical analysis, VEGF–DNP in DNP-KLH versus VEGF–DNP + αDNP Ab **$p = 0.0066$, Student's $t$-test ($n = 1$). **c** As in **a** except for using OPN–DNP. Statistical analysis, OPN–DNP in DNP-KLH versus KLH, **$p = 0.0080$, Student's $t$-test. (7 mice per group) ($n = 2$). **d** Combination immunotherapy. 4T1 tumor-bearing mice were treated with VEGF–DNP only twice to reduce the effect of monotherapy. Neoantigens were induced by systemic administration of nucleolin aptamer–Smg-1 siRNA conjugates 1 day prior to antibody coating with VEGF–DNP conjugate treatment as described in ref.[32] To recruit DC to the tumor site in advance of antibody coating, mice were treated with Flt3L starting 1 day prior to VEGF–DNP conjugate treatment (see Methods for details). Left panel, tumor volume 14 days after tumor implantation. Statistical analysis, Flt3L + VEGF–DNP versus Flt3L *$p = 0.0127$, Nuc-SMG1 + VEGF versus Nuc-SMG1 *$p = 0.0351$. (6 mice per group), Student's $t$-test ($n = 1$). Data are presented as mean ± SEM

its cytoplasmic domain (ΔPSMA) to prevent its internalization upon ligand binding[26]. Supplementary Fig. 1b shows that sera from DNP, but not control, immunized mice bound to ΔPSMA expressing B16.F10 cells in the presence of PSMA-binding aptamer, but not control aptamer, conjugated to a DNP molecule. To test whether tumor growth is affected in DNP-immune mice, ΔPSMA-expressing B16/F10 melanoma tumor cells (ΔPSMA-B16) were implanted subcutaneously or injected intravenously into C57BL/6 mice and local tumor growth or lung metastasis was monitored, respectively. Consistent with the scenario depicted in Fig. 1, only when mice were immunized against DNP (DNP-KLH), but not mock immunized (KLH) and treated with PSMA aptamer–DNP conjugate, local tumor growth (Supplementary Fig. 1c) and lung metastasis (Supplementary Fig. 1d) were significantly inhibited.

The need to identify products expressed on the surface of tumor cells that are not internalized upon ligand binding will significantly limit the clinical applicability of this approach. We have previously shown that targeting 4-1BB costimulation to tumor via VEGF or osteopontin (OPN) in mice resulted in effective antitumor immunity in multiple murine tumor models[27], and that combination with radiation further enhanced the scope of VEGF-targeted 4-1BB costimulation to tumors that otherwise did not express or expressed low levels of VEGF. Thus tumor-secreted products like VEGF or OPN could serve as broadly useful targets to deliver (immune) therapeutics to all type of tumors regardless of their origin. Whereas both VEGF and

OPN are secreted into the extracellular matrix (ECM) of tumors, a significant proportion of the tumor-secreted VEGF remains associated with the tumor on its surface, presumably through low affinity binding to heparan sulfate on proteoglycan expressed by tumor cells[28–30]. The physiological importance of cell-associated VEGF was suggested by the observation that in colorectal cancer patients tumor-associated VEGF correlated with disease progression[31]. We thus tested whether the DNP hapten can be targeted to the tumors by conjugation to either VEGF-binding or OPN-binding oligonucleotide aptamers.

The DNP hapten was hybridized to a VEGF or OPN aptamer via complementary sequences as described in ref.[27] To determine whether tumor growth can be inhibited in DNP-immune mice treated with VEGF–DNP or OPN–DNP conjugates, Balb/c mice immunized with DNP-KLH or with KLH were implanted subcutaneously with 4T1 breast carcinoma tumor cells and 7–8 days later when tumors became palpable mice were treated with VEGF–DNP or OPN–DNP conjugates. As shown in Fig. 2a and b, the VEGF, but not control, aptamer-targeted DNP conjugate inhibited tumor growth in DNP immune (DNP-KLH), but not in mock (KLH)-vaccinated mice. Consistent with the hypothesis that polyclonal antibodies are more effective in eliciting tumor immunity, a monoclonal anti-DNP antibody was largely ineffective (Fig. 2b). Similarly, an OPN-targeted DNP conjugate also inhibited tumor growth in the DNP-vaccinated mice. These experiments also show that clearance of the VEGF aptamer–DNP immunocomplexes by the reticuloendothelial

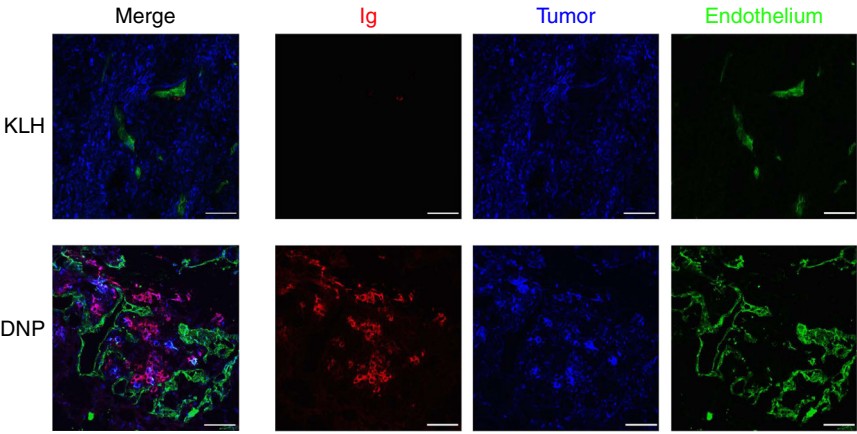

**Fig. 3** Immunoglobulin deposit formation in the tumors of DNP, but not KLH, immunized mice treated with VEGF–DNP. Palpable 4T1 tumor-bearing mice prevaccinated against DNP or KLH were injected intravenously with VEGF aptamer–DNP conjugate and 24 h later tumors were excised. Immunohistochemistry was performed using antibodies against IgM/IgG (red), pan-endothelial marker (green), and wide spectrum cytokeratin staining mainly the 4T1 tumor cells (blue). Tissue sections were imaged using a Leica SP5 inverted confocal microscope with a PL APO CS ×20/0.70 objective as described. Scale bar = 50 μm. ($n = 1$)

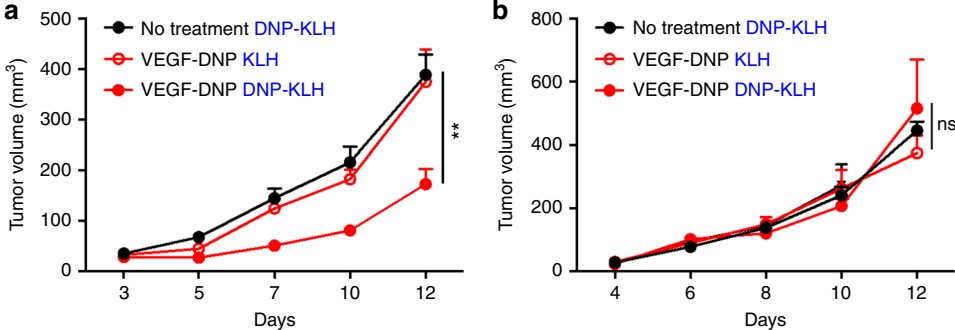

**Fig. 4** FcR expression dependence of VEGF–DNP-mediated tumor inhibition in DNP immune mice. Wild type (**a**) and FcR-deficient (**b**) C57BL/6 mice were immunized against DNP (DNP-KLH) or mock immunized (KLH), challenged subcutaneously with MC38 tumors and 4–5 days later when tumors became palpable mice were treated with VEGF–DNP conjugate and tumor growth was measured. Statistical analysis (**a**), VEGF–DNP in DNP-KLH versus KLH vaccinated mice, **$p = 0.0025$, student's $t$-test (7 mice per group) ($n = 1$); ns not significant ($p > 0.05$). Data are represented as mean ± SEM

system prior to reaching the tumor is not a limiting factor. As we have previously shown[27], the importance of VEGF and OPN tumor targeting is demonstrated by the fact that VEGF–DNP and OPN–DNP treatment was ineffective against 4T07 tumors, which are closely related to 4T1 tumors but express low levels of VEGF or OPN (Supplementary Fig. 2)

It is reasonable to assume that the antibody recruitment approach, or for that matter any immune-based cancer therapy, will not be curative as monotherapy. We, therefore, assessed whether combination with potentially complementary immune treatments could further augment VEGF–DNP-mediated tumor inhibition. We hypothesized that increasing the antigenic content of the tumor cells or recruiting more DC to the tumor will enhance tumor inhibition. Using an approach we have previously described, neoantigens were induced in tumor cells by inhibition of the nonsense-mediated mRNA decay (NMD) process using siRNAs to downregulate Smg-1, a key mediator of the NMD pathway that was targeted to the tumors in situ by conjugation to a PSMA-binding aptamer[32]. In this study, the Smg1 siRNA was targeted to tumors using a nucleolin-binding aptamer. Nucleolin is translocated to the cell surface of most tumor cells of both murine and human origin[33], and hence represents a broadly useful target to deliver therapeutics to tumor cells in vivo (also see below). To increase the number of tumor-resident cross-priming Batf3 + CD103 + DC, mice were treated with Flt-3 ligand (Flt3L), as previously described[34–36]. As shown in Fig. 2d, both

neoantigen induction in the tumor cells and Flt3L recruitment of DC when combined with VEGF–DNP treatment enhanced tumor inhibition.

**T cell-dependent hapten-mediated tumor inhibition.** The tumor inhibition experiments shown in Fig. 2 are consistent with the scenario depicted in Fig. 1, whereby coating tumor cells with DNP attracts DNP-specific polyclonal antibodies to the tumor that promote their FcR-mediated uptake by dendritic cells leading to induction of antitumor immunity. In agreement with this mechanism, immunohistochemical analysis has shown that tumors from DNP immune mice, but not control immunized mice, treated with VEGF–DNP are infiltrated with antibodies that are closely associated with tumor cells and adjacent ECM but not with endothelial cells (Fig. 3 and Supplementary Fig. 3), consistent with the cell-associated nature of tumor-secreted VEGF as discussed above.

To determine whether the observed tumor inhibition in DNP-immune mice treated with VEGF–DNP is FcR dependent, wild-type, and FcR-deficient C57BL/6 mice were vaccinated against DNP, challenged with MC38 tumor cells, and treated with VEGF–DNP. As shown in Fig. 4, whereas tumor growth was inhibited in DNP immune wild-type mice treated with VEGF–DNP (Fig. 4a), tumors grew in the FcR-deficient mice (Fig. 4b). This experiment also extends this approach to a second

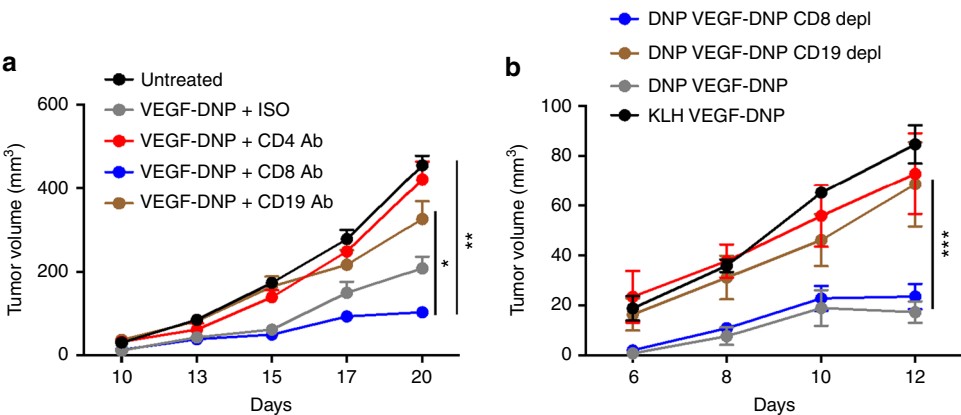

**Fig. 5** Role of immune subsets in VEGF–DNP conjugate-mediated tumor inhibition. **a** DNP immune mice were implanted subcutaneously with 4T1 tumors and treated with VEGF–DNP conjugates as described in Fig. 2. As indicated, mice were treated with depleting anti-CD4, anti-CD8, anti-CD19 and isotype control antibodies, and monitored for tumor growth. CD4 Ab versus isotype control, **$p = 0.006$; CD19 Ab versus isotype Ab, *$p = 0.022$, Student's $t$-test. (7 mice per group) ($n = 2$). **b** Splenocytes from DNP immune (DNP) mice shown in **a** and from mock immunized mice (KLH) treated with VEGF–DNP were transferred to naive Balb/c mice, challenged with 4T1 tumor subcutaneously and tumor growth was measured. CD8 depletion versus KLH + VEGF–DNP, ***$p = 0.0002$, Student's $t$-test. (7 mice per group) ($n = 2$). Data are represented as mean ± SEM

murine tumor model and underscores the broad applicability of VEGF-targeted therapy as previously described[27].

Antibody depletion experiments shown in Fig. 5a, suggests that the VEGF–DNP-mediated antitumor response was mediated by CD4+, but not CD8 + T cells. This was confirmed in an adoptive transfer experiment shown in Fig. 5b in which splenocytes from mice treated as shown in Fig. 5a were transferred to naive mice and then challenged with tumor. CD19 antibody depletion partially abrogated tumor growth (Fig. 5a) whereas adoptive transfer of splenocytes from CD19 Ab-treated mice was more effective (Fig. 5b). Thus, antitumor immunity appears to be at least partially dependent on CD19 expressing, presumably B, cells. Consistent with these observations, antitumor immunity mediated by coating tumor cells in situ with allogeneic antibodies was also dominated by CD4 + T cells[37], and tumor inhibition in mice with orthotopically established glioma treated with tumor lysate and OX40-Fc fusion protein was CD4+, but not CD8, T cell-dependent, and was also B cell-dependent[38].

The tumor-infiltrating lymphocytes analyzed by multiparameter flow cytometry 2 days following the last treatment of DNP immune or mock immunized mice with VEGF–DNP exhibited a proinflammatory Th-1 signature (Supplementary Fig. 4). There was a relative increase in non Treg CD4+, but not CD8+, T cells, consistent with the CD4 but not CD8 dependence of the antitumor response shown in Fig. 5, and absolute Treg numbers decreased while the CD8/Treg and CD4/Treg ratios increased. The numbers of tumor-infiltrating MHC class II$^{high}$ expressing macrophages, M1 macrophages, and DC increased whereas the numbers of MHC class II$^{low}$ and M2 macrophages decreased.

**Recruiting anti-αGal antibodies to tumors.** The protocol to coat tumor cells in situ with endogenous polyclonal antibodies as depicted in Fig. 1 consists of two steps, vaccination against an antigen/hapten and treatment with a tumor-targeted said antigen/hapten. While the vaccination step is simple and carries low risk, it should be possible to dispense with vaccination altogether by recruiting naturally occurring antibodies. Natural antibodies against the trisaccharide epitope Galα1-3Galβ1-4GlcNAc-R (αGal) are the most abundant antibody (1% of total immunoglobulin) in the sera of humans, produced throughout life as a result of constant antigenic stimulation by carbohydrate antigens on commensal bacteria in the GI tract[39]. Wild-type mice do not

produce anti-αGal antibodies, since it is synthesized in mice and therefore it is a self-antigen to which mice are tolerant[39]. To model the recruitment of natural antibodies in mice, we used an αGal-deficient mouse strain that was generated by deleting the α1,3 galactose transferase, a key enzyme in the synthesis of αGal (GT KO mice)[40]. While αGal antibody titers in the GT KO mice housed in the vivarium with reduced exposure to commensal bacteria is low, immunization with xenograft tissue such as rabbit red blood cells or pig kidney membrane homogenate induces high levels of anti-αGal antibodies comparable to that found in human serum[41]. Unlike human tumors that are αGal-null, mouse tumors will express αGal-containing glycoproteins and thereby will be rejected in the GT KO mice immunized against αGal. Since B16BL6 tumors, a highly metastatic subline of B16 tumors, has downregulated GT and does not express αGal, it has been used as a suitable model for human αGal-null tumors[42,43].

The GT KO mice are of mixed genetic background (C57BL/6xDBA/2Jx129sv GT−/−) because homozygous GT KO H-2b mice do not breed well and produce reduced anti-αGal titers (Uri Galili, personal communication). The low titers of αGal antibodies in the GT KO mice were boosted by immunization with pig kidney membrane homogenates. To test whether tumor recruitment of the αGal antibodies can inhibit tumor growth, mice implanted with B16BL6 tumors were treated with VEGF-targeted αGal trisaccharide. B16BL6 tumor cells, derived from the metastatic B16.F10 melanoma tumor cell line[44], grow in the GT KO mice except that one has to inject ~$5 × 10^5$ cells[45]. As shown in Fig. 6, tumors failed to grow in the VEGF-αGal-treated mice, 5/6 mice remaining tumor free for the duration of the experiment, provided the mice were boosted with the kidney membrane homogenates. Importantly, vaccinated mice treated with VEGF–DNP conjugates did not reject the B16BL6 tumors. This experiment, therefore, shows that tumor rejection was dependent on the endogenous αGal antibodies and their recruitment to the tumor.

To assess the potential relevance to human patients we examined whether VEGF aptamer-targeted αgal hapten can recruit antibodies present in the human serum to tumors, but not to normal tissue. Frozen sections of tumor and matched normal tissue from the same patient were first incubated with VEGF aptamer–αgal conjugate or with VEGF aptamer–DNP conjugate used as a control, and then incubated with human serum. As

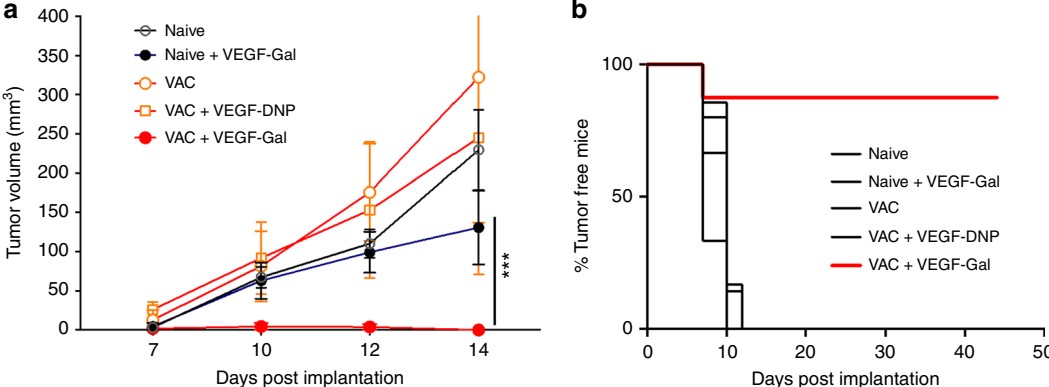

**Fig. 6** Targeting αGal trisaccharide to tumor cells. αGal-null (C57BL/6xDBA/2Jx129sv GT−/−) mice were vaccinated with pig kidney extracts to boost the titer of anti-αGal antibodies and 10 day later challenged subcutaneously with $5 \times 10^5$ αGal$^{low/negative}$ B16BL6 tumor cells. 3 days post tumor implantation mice were treated with VEGF–DNP or VEGF–αGal hapten conjugates and tumor growth was measured. **a** Tumor volume. Statistical analysis: VEGF–αGal in VAC versus naive, ***$p = 0.0037$, Student's $t$-test. **b** Time to tumor appearance. Statistical analysis: VEGF–αGal in VAC versus naive or any other group, $p < 0.0001$, Log-rank test for significance. (7 mice per group) ($n = 2$). Data are represented as mean ± SEM

shown in a representative example in Fig. 7a, human serum bound to a tumor section obtained from a kidney tumor biopsy, but not to normal tissue from the same individual, that was incubated with VEGF-αgal but not with VEGF–DNP conjugates. Figure 7b shows that four sera obtained from healthy donors recruited antibodies to tumor, consistent with the prevalence of anti-αgal antibodies in the human population. Figure 7c shows that VEGF-αgal, but not VEGF–DNP, conjugate recruited antibodies to tumors of endometrial, kidney, melanoma, and colon origin, altogether 5 out of 5 tumors tested, reflecting the broad expression pattern of VEGF in tumors and the utility of VEGF aptamer targeting ligand, as we have previously demonstrated in mice[27,46]. The variation in staining intensities seen in Fig. 7b and c may reflect variation in the titer of anti-αGal antibodies or intratumoral VEGF expression, respectively. Taken together, the experiments shown in Fig. 7, supports the view that treatment with VEGF aptamer-targeted αgal conjugate will be capable of recruiting polyclonal antibodies to the tumors of most if not all cancer patients.

## Discussion

In this study, we describe a novel approach to enhance the FcR dependent uptake of tumor cells in situ by coating tumor cells with endogenous polyclonal antibodies, leading to potent T cell-mediated antitumor immunity. The approach was first to vaccinate mice against DNP, an innocuous hapten, establish tumors in the mice, and then treat the vaccinated tumor-bearing mice with a VEGF or OPN aptamer–DNP conjugate that target the DNP hapten to the tumor and attracts the preformed polyclonal DNP antibodies (Fig.1). In several murine tumor models, the H-2d Balb/c-derived 4T1 breast carcinoma (Fig. 2) and H-2b C57BL6 MC38 colorectal cancer (Fig. 4a) models, this led to a significant inhibition of tumor growth. DNP vaccination prior to tumor establishment is not mandated by this approach; rather it is necessitated due to the rapid growth of the transplantable tumors in mice. Conceivably in human patients where tumors grow more slowly vaccination of patients with minimal or low tumor burden will provide sufficient time to induce an immune response against the hapten or antigen.

The 4T1 breast carcinoma tumor cell line is a poorly immunogenic tumor and most immune-based monotherapies are ineffective once the tumor becomes palpable[47]. Considering that the treatment conditions used in our studies are probably suboptimal in terms of the aptamer–hapten conjugate and treatment

regimen, that monotherapy with VEGF or OPN-targeted DNP conjugates was capable of inhibiting the growth of palpable 4T1 or MC38 tumors suggest that the polyclonal antibody-coating recruitment approach could be a potent way of stimulating antitumor immunity. This is also supported by the experiment showing that coating tumor cells with naturally occurring anti-αgal antibodies prevented the outgrowth of B16BL6 tumors in the majority of the treated mice (Fig. 6). On note, B16BL6 is a highly aggressive tumor cell line derived from the already metastatic B16.F10 melanoma tumor cell line by selection for enhanced tissue invasiveness[44].

We speculate that the polyclonal nature of the tumor recruited (DNP-specific) antibodies providing synergy among diverse Fc-encoded effector functions[14,19,20,48] contributed to the effectiveness of the inhibition (Figs. 2, 4a and 6), and may have been also the reason for the remarkable antitumor immune responses seen in a minor MHC mismatched tumor model[25]. Consistent with this hypothesis, in a phase II/III clinical trial of patients with malignant ascites treated with Catumaxomab, a rat/murine anti-CD3 and EpCAM monoclonal antibody, induction of (polyclonal) human anti-mouse antibodies correlated with clinical benefit[49]. An alternative to the polyclonal antibody-recruiting approach is to use monoclonal antibodies. How effective are monoclonal antibodies targeting a specific product on tumors in terms of inducing T cell immunity (i.e., vaccinal effect) is not clear. We are aware of three studies demonstrating a vaccinal effect in mice treated with monoclonal antibodies[9,50,51]. Though hard to judge, considering the tumor models and/or experimental conditions used, the antitumor effects seen on those studies appear to have been modest compared to what was seen in this study. Indeed, in our study treatment with a monoclonal anti-DNP antibody was largely ineffective (Fig. 2b). The concept of recruiting antibodies to cell surface with ligand targeted antigens has been previously described[21–24]. Here we describe a broadly applicable, if not universal, drug formulation to coat tumor cells in vivo with polyclonal antibodies whereby naturally occurring ubiquitously present antibodies are recruited to tumor cells in mice using a pan-cancer VEGF targeting ligand conjugated to an aGAL trisaccharide, and show that it leads to the regression of highly aggressive B16BL6 tumors (Figure 6).

We have not observed overt toxicity in DNP-vaccinated mice treated with VEGF–DNP in terms of mortality, morbidity, or weight loss. Nonetheless, we cannot exclude that subclinical levels of antibody–antigen aggregates have formed that may become clinically important upon intensification of therapy. In that case,

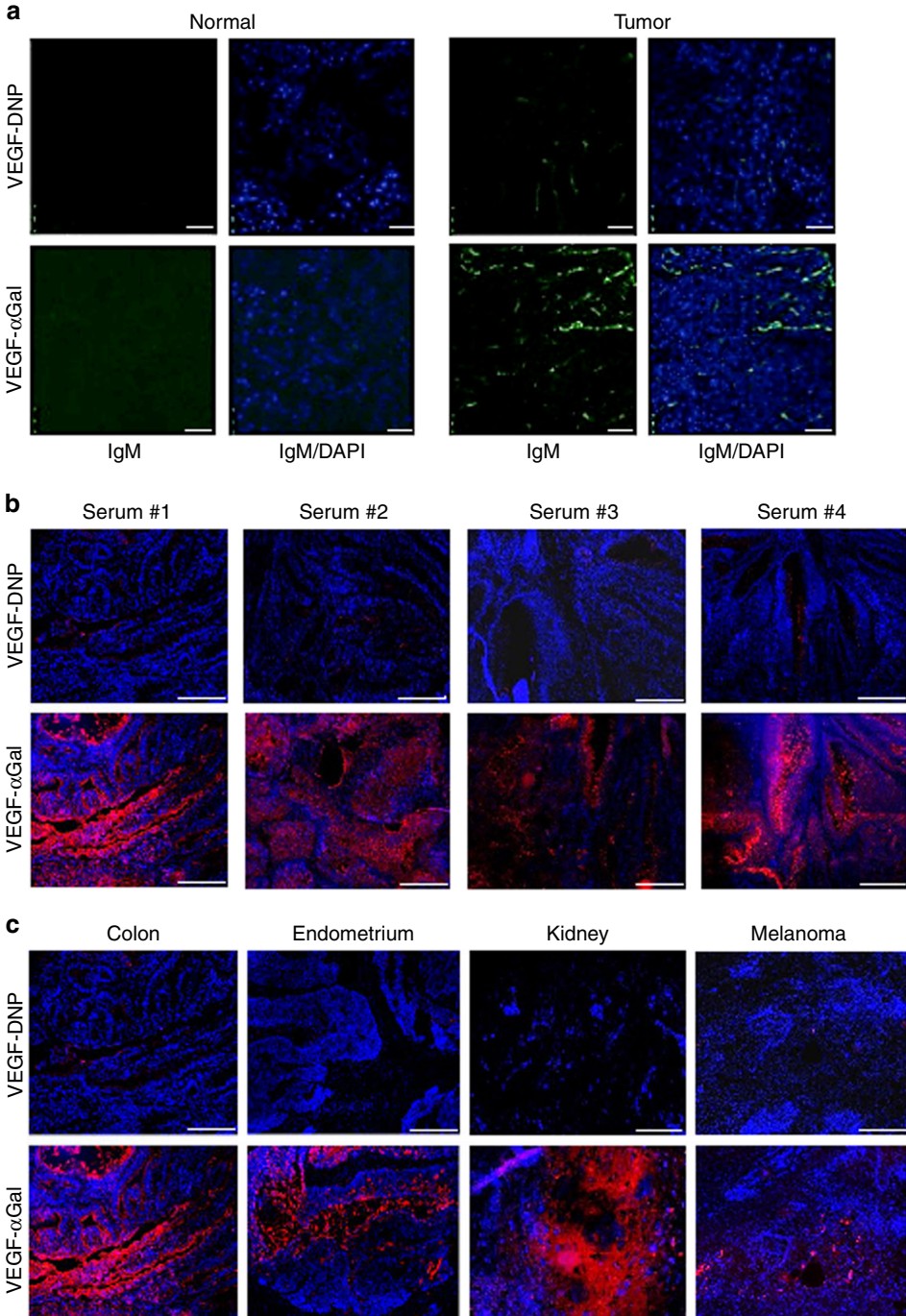

**Fig. 7** VEGF–αGal mediated recruitment of antibodies from human serum to tumor biopsies. Frozen section of normal and tumor biopsies were first incubated with VEGF–DNP or VEGF–αGal, then with human serum and stained with either DAPI (blue) or with mouse anti-human IgM-Alexa488 A (green) and visualized by confocal microscopy at ×40 magnification. **a** Renal cell cancer (RCC) and matched normal tissue. **b** Colon biopsy was incubated with VEGF–αGal or VEGF–DNP and three additional human sera. **c** Tumor biopsies from colon (shown in **b**), endometrium, kidney (different from **a**), and melanoma were incubated with VEGF–αGal or VEGF–DNP conjugate and the human serum used in **a**. Scale bar = 25 μm. (n = 2)

encapsulation of the aptamer-targeted hapten conjugates into biodegradable nanoparticles that would preferentially accumulate at the tumor site via the EPR effect could be considered.

Combination with complementary forms of immune therapy such as increasing the neoantigen burden of the tumor cells or recruiting more dendritic cells to the tumor lesion further enhanced tumor inhibition (Fig. 2d), suggesting that the antibody recruitment approach could represent a general strategy to

enhance immune susceptibility of tumors and potentiate any form of immune therapy.

In this study, we used oligonucleotide aptamers to target haptens to the tumor. Aptamers can be readily synthesized in a cell-free chemical reaction, and conjugation to the therapeutic cargo involves a simple hybridization between complementary sequences engineered at the end of the aptamer and cargo[52]. Nonetheless, generating aptamers with sufficient avidity for use as

targeting ligands is challenging. Other forms of targeting ligands that can be considered for this purpose are monoclonal antibodies or peptides. Monoclonal antibodies exhibit high avidity to their targets, yet since antibodies are cell-based products and attachment of a therapeutic cargo requires chemical conjugation, their use as targeting ligands is less straightforward. While a large library of peptide ligands is available, generated mostly through phage display, their invariably low avidity precludes their use as targeting ligands unless used in multimeric form.

The DNP hapten was targeted to VEGF and OPN, two products that are secreted into the stroma of many tumors and can be further upregulated by radiation[27,46]. Tumor cell secreted VEGF remains largely associated with the tumor cells through low avidity binding to heparan sulfate moieties on proteoglycans secreted by tumor cells[28–30], and the secreted OPN may bind to its integrin and CD44 receptors expressed on the tumor cell[53]. Consistent with this, immunohistochemical analysis (Fig. 3 and Supplementary Fig. 3) has shown that immunoglobulins accumulate in the tumors of DNP immune mice treated with VEGF–DNP conjugate and they are found mostly in close proximity to the tumor cells, but not the tumor vasculature. Targeting to VEGF or OPN was used to demonstrate the concept of antibody recruiting and may not be the optimal targets for tumor delivery of hapten/antigens. Future studies could explore other products secreted into the tumor stroma like metalloproteases or TGFβ, or cell surface products expressed on tumor cells like Her2, PSMA, EpCAM, or nucleolin.

The mechanism depicted in Fig. 1 is further supported by experiments showing that tumor inhibition in DNP-immune mice treated with VEGF–DNP was accompanied by intratumoral accumulation of immune cells with a Th1 proinflammatory signature (Supplementary Fig. 4), was FcR dependent (Fig. 4) and T cell mediated (Fig. 5). It is noteworthy that tumor inhibition was dependent on CD4+, but not CD8+, T cells (Fig. 5). This is consistent with previous studies showing that in situ-coating tumor cells with monoclonal[50,51] or allogeneic[37] antibodies elicited a CD4+ dominant T cell response. Tumor inhibition was also dependent on CD19, presumably B, cells (Fig. 5), which is consistent with another study showing that a CD4+ T cell-driven antitumor response was also dependent on CD19 cells[38].

The treatment protocol depicted in Fig. 1 is a two-step procedure, vaccination against a hapten/antigen and treatment with a ligand-targeted hapten/antigen. Both the vaccine and the ligand–hapten/antigen conjugate are simple inexpensive chemically synthesized reagents that would be broadly if not universally applicable to all cancer patients. For example, most tumors can be targeted via VEGF, which is expressed in many tumors or can be upregulated by irradiation[27,46], and illustrated in this study (Fig. 7c). Whereas vaccination against a hapten or antigen is a simple and low risk procedure it could be dispensed altogether by recruiting naturally occurring antibodies like the highly abundant αGal-specific antibodies that constitute about 1% of the total circulating immunoglobulin in the human serum[39]. Using a murine model that simulates the scenario in human patients, we have shown that treatment of the tumor-bearing mice with VEGF aptamer-targeted αGal hapten prevented tumor growth (Fig. 6), and recruited antibodies from 4/4 human sera tested (Fig. 7b) to 5/5 human tumor sections of distinct origin (Fig. 7c). Taken together, these experiments suggest that treatment with a systemically administered, chemically synthesized, broadly applicable drug could recruit pre-existing antibodies and sensitize any tumor lesion in every cancer patients to the immune system and thereby contribute to control of tumor growth, especially when used in combination with other immune potentiating treatments as shown in Fig. 2d.

## Methods

**Construction of aptamer conjugates**. A chemically synthesized VEGF aptamer[54], OPN aptamer[55], or A10 PSMA aptamer[56] with a 21 nt extension 5′-GUUCU-CAUGCACACUUAUAGC-3′ was annealed to a chemically synthesized complementary oligonucleotide containing a 5′ DNP (Trilink) or αGal hapten. An azide-modified complementary oligonucleotide was conjugated to propargyl-modified αGal hapten (Elicityl) by click chemistry. The propargyl-αGal hapten was incubated with the azide-oligonucleotide (2:1) in the presence of 10 mM TBTA-$Cu^{2+}$ complex (Lumiprobe), 10 mM ascorbic acid, 0.1 M triethylammonium acetate, and 75% DMSO. Reaction was rotated for 2 h at room temperature and then purified by polyacrylamide gel electrophoresis. Control aptamer that does not bind to VEGF described in reference[5] (aptamer ARC225) has the same sequence as the VEGF-binding aptamer except for 7 substitutions of 2′-O-methyl with hydroxyl residues. Equimolar amounts of aptamer and extended DNP or αGal were mixed, heated to 82 °C and cooled to room temperature. Annealing efficiency, monitored by agarose gel electrophoresis, was >90%.

**Vaccination protocol**. The facilities at the University of Miami's Division of Veterinary Resources are fully accredited by the Association for Assessment and Accreditation of Laboratory Animal Care and U.S. Drug Administration. An Office of Laboratory of Animal Welfare assurance is on file, ensuring that humane animal care and use practices, as outlined in the Guide for the Care and Use of Laboratory Animals, are followed. 5- to 6-week-old Balb/c (H-2d) and C57BL6 (H-2b) mice were purchased from The Jackson Laboratory. FcγR knockout mice (C57BL6 background) were purchased from Taconic. For vaccination, 100 μg DNP-KLH or KLH emulsified with PBS and IFA was administered to mice i.p. three times every 2 weeks. 2 weeks after the last boost, mice were injected with tumor cells.

**4T1 and MC38 tumor models**. *The 4T1, 4T07 and MC38 tumor cell lines were obtained from the ATCC and NCI: DCTD tumor/cell line repository, respectively,* and each cell line was confirmed to be mycoplasma-free prior to use in experiments. Balb/c or C57BL6 mice were injected subcutaneously on the right flanks with $2.0 \times 10^4$ 4T1 or $5.0 \times 10^5$ MC38 tumors cells, respectively. Tumors became palpable about days 8–9 with 4T1 or day 4–5 with MC38, at which point treatment with VEGF–DNP conjugates was given every third day for a total of three treatments at 500 pmol per mouse per injection administered i.v. Alternatively, the VEGF–DNP hapten conjugate was incubated with a monoclonal anti-DNP or isotype control antibody (D8406, SIGMA-Aldrich) at a ratio of 2:1 for 20 min at 37 °C and injected i.v. into non immune mice. 1.0 nmole of Nucleolin–Smg1 conjugated was administered i.v. three times 3 days apart. 20 mg FLT3L (Bioxcell #BE0098) was given i.p. every other day starting 2 days after tumor injection until the last i.p. injection of VEGF–DNP for a total of 6 injections. For depletion studies, mice were given anti-CD4 (Bioxcell #BE0003-1), anti-CD8 (Bioxcell #BE0004-1) antibodies (100 μg/mouse, once weekly), or anti-CD19 antibody (Bioxcell #BE0150) (300 μg/mouse, every 4 days) starting 1 week prior to tumor injection which continued throughout the experiment. 2 days after the last VEGF–DNP injection, spleens from mice were collected and dissociated into single-cell suspensions. Splenocytes ($1.5 \times 10^7$) and 4T1 ($2.0 \times 10^4$) were injected i.v. and s.c., respectively, on the same day into naive Balb/c mice.

**B16-ΔcdPSMA immunotherapy**. C57BL6 mice (Jackson Labs) were injected with $5.0 \times 10^5$ B16.F10 cells either s.c. or i.v, as indicated. 5 days after injection, mice were treated with PSMA–DNP conjugates every third day for a total of three injections and tumor volume was monitored. For metastasis study, when control mice started showing signs of morbidity, all mice were killed and lungs were weighed. B16.F10 cells were obtained from the MD Anderson cell repository and confirmed to be mycoplasma-free prior to injection into the mice.

**αGal experiments**. The trisaccharide epitope Galα1-3Galβ1-4GlcNAc-R (αGal) containing an alkyne group (Elicityl) was conjugated to an azide-modified oligonucleotide as described for DNP using $Cu^{++}$-free click chemistry and purified by HPLC. The ODN–αGal conjugate was annealed to VEGF aptamer as described above. αGal-null mice (obtained from Kim Wigglesworth, University of Massachusetts, Worcester) were vaccinated with pig kidney extracts as previously described[57]. 2 weeks after vaccination, mice were injected s.c. with $5 \times 10^5$ αGal$^{low/null}$ B16BL6 tumor cells. Treatment via i.p. injection was started on day 3 post tumor administration, 1 day before tumors became palpable, and given every other day for a total of three injections. B16BL6 cells were obtained from the MD Anderson cell repository and confirmed to be mycoplasma-free prior to injection into the mice.

**Dissociation of tumor and FACS analysis**. Tumors were harvested 2 days after last treatment and weighed. Cells were isolated by dissecting tumor tissue into small fragments followed by digesting in 1 mg/ml of collagenase in complete RPMI media (10% FBS, 1% penicillin, streptomycin) prior to using Gentle MACS Dissociator. Cell suspension was passed through a 70 μm nylon strainer to obtain a single-cell population. Cells were then incubated with fixable viability dye eFluor780 for 10 min at 4 °C and washed twice in FACS buffer prior to staining in order distinguish live cells from the dead cells. Cells were incubated with Fc

blocking antibodies for 10 min at 4 °C, then incubated for 30 min at 4 °C with antibodies from Biolegend, CD45-FITC(#103107), CD19-APC Cy7(#115529), CD3-APC Cy 7(#100221), CD11c-PE Cy7(#117317), MHCII-BV785(#107645), CD8a-BV510(#100751), CD103-BV421(#121421), CD11b-AF710(#101222), Ly6C-Percp Cy5.5(#128011), Ly6G-BV605(#127639), F4/80-APC(#123115), PD1-BV421 (#135217), Tim3-APC(#134007), CD4-A700(#100429), CD25-PE Cy7(#101915), CD335-BV605 (BD Biosciences #564069), and L/D-eFluor780 (Thermo #65-0865-14). Antibodies were used at concentration of 0.25 µg/100 µl.

For intracellular staining, cells were fixed and permeabilized with FOXp3 staining buffer kit (eBioscience #00-5523-00) according to manufacturer's instructions. Cells were incubated for 30 min at room temperature in dark with FOXP3 Ab-PE at a concentration of 0.25 µg/100 µl (Biolegend #126403)and analyzed using the CytoFLex (BD Biosciences) and Kaluza software (Beckman Coulter).

**Immunohistochemistry**. 4T1 subcutaneously established tumors were removed and snap-frozen in OCT, sectioned (UMiami Diabetes Research Institute Histology Core) and stained. Briefly, slides were warmed to room temperature followed by a short incubation in acetone. Slides were washed with PBS and blocked for 1 h with 3% BSA in PBS with 0.01% Tween-20. Slides were stained overnight at 4 °C in block buffer with anti-VEGF Ab (R&D Systems #AF-493-SP), a wide spectrum anti-cytokeratin(Abcam #ab9377) and anti-IgM/IgG (H + L) antibodies (Jackson ImmunoResearch #115-116-068), anti-IgM-PE (Jackson ImmunoResearch #115-116-075) and IgG-PE (Jackson ImmunoResearch #115-116-146), or pan-endothelial cell stain (Biolegend #120505). Antibodies were used at concentration of 0.25 µg/100 µl. Slides were then washed and incubated with the appropriate secondary antibodies in block buffer for 1 h at room temperature. Slides were washed and coverslip were mounted using Prolong Gold Antifade (ThermoFisher Scientific). Image acquisition was performed at the Analytical Imaging core at DRI/SCCC, University of Miami using a Leica SP5 inverted confocal microscope equipped with ×20 PL APO CS/0.70 and ×40 HCX PL APO CS/1.25

WT or CT26 cells stably expressing a mutant form of human PSMA with a cytoplasmic deletion[26] were cultured in RPMI. PSMA–DNP aptamer was incubated with cells for 30 min at 37 °C followed by incubation with rabbit anti-DNP-Alexa488 Ab (ThermoFisher Scientific #A-11097). Cells were then fixed in PFA at RT for 10 min and mounted with Prolong Gold anti-fade as described above. CT26 cell line was obtained from ATCC and was confirmed to be mycoplasma-free prior to injection into the mice.

Sections from frozen tumor and matched normal tissue were obtained from the University of Miami Tissue Bank were prepared as described above. Sections were first incubated with 100 pmol VEGF aptamer conjugate for 1 h, washed, followed by incubation of 1:40 dilution of normal de-identified human serum obtained from a commercial vendor, washed and then incubated with mouse anti-human IgM-Alexafluor 488 (Biolegend #314533). After washing, coverslips were mounted using Prolong Gold Antifade with DAPI and imaged by confocal microscopy were collected using the Microbiology and Immunology Keyence BZ-X710 equipped with Nikon objectives ×10 PlanApo NA 0.45 and ×20 PlanApo NA 0.75.

**Statistical analysis**. All statistical analyses were performed using GraphPad prism software. For tumor volume comparisons and TIL analysis parametrical statistical methods (one-way ANOVA with Tukey post test) were used. Log-rank tests were used for Kaplan–Meier curves of survival. Mann–Whitney $U$ test was used to compare variables between two groups. When three or more groups were compared, the Kruskal–Wallis test with Dunn posttest was used. All statistical analyses were performed using GraphPad software. Data were considered significant when $P < 0.05$. Notation in figures: $^{ns}p > 0.05$; $*p = 0.05$–$0.01$; $**p = 0.01$–$0.001$; $***p < 0.001$.

**Study approval**. Experiments with mice were approved by the University of Miami institutional IACUC. An exemption from IRB approval was received from the University of Miami to conduct the experiments with de-identified human tumor biopsies obtained from the local tumor bank.

**Data availability**. The authors declare that the data supporting the findings of this study are available within the paper and its supplementary information files and/or from the corresponding author upon reasonable request.

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

## Acknowledgements

We thank Uri Galili and Kim Wigglesworth for providing the GT KO mice and help in establishing the experimental system, Oliver Umlaud for assisting with flow cytometry, Kevin Johnson for assisting with immunohistochemistry and Tan Ince for providing human tumor samples. This work was supported by the Dodson foundation and the Sylvester Comprehensive Cancer Center, Medical School, University of Miami. This work was supported by the Dodson foundation and the Sylvester Comprehensive Cancer Center, Medical School, University of Miami.

## Author contributions

B.S. was responsible for the murine immunotherapy, immunological studies, and helped write the manuscript; E.C. was responsible for the immunohistochemistry experiments; AL was responsible for the splenocyte transfer experiment, established the GT KO colony and was responsible for the immunotherapy experiment; A.R.C. was responsible for generating the VEGF–DNP conjugates; R.B. was responsible for generating the PSMA–DNP reagent and carrying out the experimental metastasis immunotherapy experiment in the ΔPSMA-B16 mice. I.C. was responsible for developing and optimizing the conditions for DNP vaccination. O.M. synthetized the ODN-αGal for hybridization to the VEGF aptamer. E.G. was responsible for designing the overall approach, coordinating the various experiments, help trouble-shoot, and write the manuscript.

## Additional information

**Competing interests:** E.G. is a founder and equity holder of Sebastian Biopharma that holds the intellectual rights for the hapten-mediated antibody recruitment approach. The remaining authors declare no competing interests.

