## [Peer Review File · Nature Communications]

Reviewers' comments:

Reviewer #1 (Remarks to the Author):

The study "Hapten-mediated recruitment of polyclonal antibodies to tumors engenders antitumor immunity" by Gilboa and colleagues introduces an elegant approach for cancer-immunotherapy by redirecting hapten specific antibodies via hapten-conjugated tumor targeting molecules. The authors introduce two variations of the approach either exploiting hapten-specific antibody responses induced by pre-vaccination of animals or relying on pre-formed natural anti-Gal antibodies. The study is well performed with diligent evaluation of the concept and effector mechanisms. Particularly, the authors invested substantial work to prove the targeting of anti-GAL antibodies to murine tumors a difficult task given the fact (i) that natural anti-GAL antibodies due to the presence of the epitope are lacking in WT mice and that murine tumors do express the tri-saccharide epitope. The proposed approach is very interesting and experimental validation is well performed and convincing. I would like suggest that the authors consider one additional study to prove the particular value of their approach. The authors argue that their approach allows targeting of polyspecific antibodies to tumors which might have a stronger antitumoral potency as compared to monoclonal antibodies. One informative control would be to evaluate this in vivo by combining DNP-VEGF treatment with application of monoclonal anti-DNP antibodies. Alternatively, the authors may consider to use monoclonal anti-GAL antibodies in their second model. This would allow a direct comparison of the potency of polyclonal vs. monoclonal antibody targeting and thereby possibly strengthen the impact of the paper. Moreover, I would like to ask the authors to add a passage into the discussion about potential risks associated with the clinical application of the approach e.g. anaphylaxis due to aggregate formation of hapten-linked tumor targeting molecules and mitigation strategies to circumvent such risks.

Reviewer #2 (Remarks to the Author):

This manuscript describes the hapten mediated recruitment of polyclonal antibodies to tumors to induce antitumor immunity. The novelty of this approach lies in using haptens targeted to tumors through tumor-binding aptamers which can be recognized by host polyclonal antibodies, either induced through immunization or through pre-existing host antibodies. The appeal of this approach lies in the potential to avoid the necessity of identifying tumor antigens suitable for targeting. The authors utilize several mouse models to demonstrate the anti-tumor potential of their approach.

There are several major and minor concerns which this reviewer feels should be addressed:

1. The authors should try to avoid overstating the implications of their study. In the final sentence of the abstract they state that, "... could enhance the susceptibility of most if not all tumours to immune elimination." I feel this sentence could be softened somewhat without losing the ability to suggest the potential utility of harnessing the power of such polyclonal host responses. This claim needs to be tempered given that all the data presented in the following results show short term tumor control which seems unlikely to be curative.

2. One issue with this manuscript is apparent novelty. The authors' themselves draw parallels in their results section that what they are investigating is, "... by-and-large recapitulating the scenario of miMHC mismatch transplanted tumor cells described by Carmi et al." Could they more clearly delineate what they are adding to this study.

3. Figure. S1. 'Targeting DNP to prostate tumor cells inhibits tumor growth in DNP immune mice'. This supplementary figure is misleading on two fronts: 1. It does not contain data from prostate tumor cells but rather PSMA expressing tumor cells. 2. More worryingly, experiments were performed in deltaPSMA-expressing B16/F10 melanoma bearing mice (Figure S1B and Fig S1C), but Figure S1A shows the PSMA binding aptamer conjugated to PSMA-DNP in deltaPSMA CT26

colon cancer tumor cells. This is not appropriate as validation and experimental should match.

4. This is a complicated manuscript with complex methodology and to aid the reader it would be highly beneficial to place statistical significance within the figure panels themselves as well as the legends. Further there is no reference to how many animals are used in experiments or whether the data is summary or representative. Nor is it detailed how many times these experiments were carried out. More detailed analysis and annotation is required throughout the figures to allow appropriate interpretation.

5. Frequently there is also a real lack of controls within the results shown. Importantly (to this reviewer) vaccinations would be better controlled if non-targeted aptamer rather than no treatment, was used in order to demonstrate specificity. I do not think it is necessary to repeat all of the work to demonstrate this but at least one or two examples in supplementary figures would put the readers' minds to rest regarding the specificity of the responses observed.

5. A real concern is caused by Figure 2. In this figure the legend states that mice were vaccinated or not and injected with 4T1 tumor cells. When tumors were palpable day 7-8 the VEGF-DNP construct was injected. If this was the case why does the VEGF-DNP arm and OPN-DNP arm show reduced growth before the reagent is administered?

6. There seems to be no real attempt to address off-target toxicity. It seems very possible that there is likely to be some binding of the aptamers to non-tumor sources of VEGF and/or OPN, and also perhaps accumulation of the haptens in the liver. The manuscript would benefit from the authors addressing this concern. Ideally data could include imaging of organs following treatment with suitably labeled versions of the aptamers or hapten specific antibodies. If the authors feel this is beyond the scope of this manuscript then they should discuss toxicity concerns in their discussion.

7. Have the authors verified and validated the binding of their aptamer conjugates by flow cytometry on the various cell lines used? Please show data.

8. Were Figure. 4A and B carried out side-by-side? In order to make this comparison they should be from the same experiment set up and presented on the same graph given that the FcR deficient mouse data are being compared to the wildtype with suitable statistical analysis.

9. Figure. 5. Do the authors have flow cytometry data demonstrating the depletion of the various lymphocyte populations? In the absence of these data it is difficult to ascertain the robustness of the lack of a role for CD8s.

10. Figure 6 is difficult to interpret given the complexity of the system used; vaccination with a pig kidney homogenates which is a complex set of foreign antigens, a complex mixed mouse strain is used and an absence of the glycosyltransferase could introduce a wide range of both potential and missing antigens outside of aGal. These issues may well be beyond the scope of this paper but there should be some discussion of these issues and further there should be a little less emphasis placed on the concept of a lack of vaccination.

REVIEWER #1 (Changes are highlighted in text and page # is indicated)

1. “One informative control would be to evaluate this in vivo by combining DNP-VEGF treatment with application of monoclonal anti-DNP antibodies”

We carried out this experiment and was added to Fig. 2 as new panel B and accompanying highlighted text, page 6 and 14. It indeed shows that a monoclonal anti-DNP antibody has, at best, a marginal effect on tumor inhibition.

2. “...add a passage into the discussion about potential risks associated with the clinical application of the approach e.g. anaphylaxis due to aggregate formation of hapten-linked tumor targeting molecules and mitigation strategies to circumvent such risks”

We added the paragraph shown below – see page 15.

“We have not observed overt toxicity in DNP vaccinated mice treated with VEGF-DNP in terms of mortality, morbidity, or weight loss. Nonetheless, we cannot exclude that subclinical levels of antibody-antigen aggregates have formed that may become clinically important upon intensification of therapy. In that case, encapsulation of the aptamer-targeted hapten conjugates into biodegradable nanoparticles that would preferentially accumulate at the tumor site via the EPR effect could be considered.”

REVIEWER #2 (Changes are highlighted in text and page # is indicated)

1. “The authors should try to avoid overstating the implications of their study. In the final sentence of the abstract they state that, ”... could enhance the susceptibility of most if not all tumours to immune elimination.” I feel this sentence could be softened somewhat ...

We changed the sentence to “... could enhance the susceptibility of ~~(most if not all)~~ a broad range of tumours to immune elimination.” Is this acceptable?

Please note that we make a point saying that the antibody recruitment approach (and for that matter any form of immune therapy) will not be curative as monotherapy, but rather contributory and hence require combinatorial approaches as illustrate din Fig. 2D.

2. One issue with this manuscript is apparent novelty. The authors’ themselves draw parallels in their results section that what they are investigating is,”.. by-and-large recapitulating the scenario of miMHC mismatch transplanted tumor cells described by Carmi et al.” Could they more clearly delineate what they are adding to this study.

The sentence is misleading and we misstated what we intended to say. The problem is with the word “recapitulate” the study by Carmi et al... Of course we are not. The cited study injected miMHC mismatched tumors in mice and observed efficient rejection of tumor that was mediated by endogenous polyclonal antibodies that “coated” the tumor cells. Hardly clinically translatable. We are describing in this manuscript a radically different, clinically useful, and potentially broadly applicable, way of “coating” tumor cells with endogenous antibodies – administering tumor-targeted haptens into mice that harbor anti-hapten polyclonal antibodies. The only parallel to the cited study is the polyclonal antibody mediated mechanism. What we meant to say that in our study we “recapitulated” (clearly a wrong term) the “coating” of tumor cells with polyclonal antibodies. In the revised manuscript we changed the corresponding reference to the previous study to reflect this point by saying “...as seen in the MHC mismatch tumor model²².” referring to the polyclonal antibodies and not to the approach as a whole. (see page 5, highlighted).

3. Figure. S1. ‘Targeting DNP to prostate tumor cells inhibits tumor growth in DNP immune mice’. This supplementary figure is misleading on two fronts: 1. It does not contain data from prostate tumor cells but rather PSMA expressing tumor cells. 2.

More worryingly, experiments were performed in deltaPSMA-expressing B16/F10 melanoma bearing mice (Figure S1B and Fig S1C), but Figure S1A shows the PSMA binding aptamer conjugated to PSMA-DNP in deltaPSMA CT26 colon cancer tumor cells. This is not appropriate as validation and experimental should match.

This is a “soft” piece of data attempting to further indicate the generality of the approach. In retrospect it may not have been wise to include this experiment for the reasons stated by the reviewer (and will be happy to remove it if the reviewer suggests).

Regarding point #1, We had to use the deltaPSMA system as described and explained in reference #23 This is an immunotherapy experiments necessitating the use of immune competent mice and syngeneic (murine) tumors and since the PSMA aptamer that is available binds to human, not murine cells, we transfected the human PSMA into the murine tumors. Reason with used deltaPSMA containing a small deletion in its amino terminus was to prevent its internalization upon PSMA aptamer engagement. Not ideal but makes the point.

Point #2, we replaced the immunohistochemistry experiment in CT-26 with a flow cytometry experiments where we staining the deltaPSMA expressing B16.F10 cells with DNP immune or control (KLH) sera in the presence of Control-DNP and PSMA-DNP, showing preferential DNP immune sera binding to cells in the presence of PSMA-DNP compared to Control-DNP (New Fig. S1B). The text (page 5) and legend was modified accordingly.

4. This is a complicated manuscript with complex methodology and to aid the reader it would be highly beneficial to place statistical significance within the figure panels themselves as well as the legends. Further there is no reference to how many animals are used in experiments or whether the data is summary or representative. Nor is it detailed how many times these experiments were carried out. More detailed analysis and annotation is required throughout the figures to allow appropriate interpretation

Was done. # animals/experiment, number of times experiment was done, and significance was also indicated in the panel (As noted in the “Methods”: $p > 0.05$ ns; $p = 0.05-0.01$ *; $p = 0.01-0.001$ **; $p < 0.001$ ***)

5. Frequently there is also a real lack of controls within the results shown. Importantly (to this reviewer) vaccinations would be better controlled if non-targeted aptamer

rather than no treatment, was used in order to demonstrate specificity. I do not think it is necessary to repeat all of the work to demonstrate this but at least one or two examples in supplementary figures would put the readers' minds to rest regarding the specificity of the responses observed.

We did that. New Fig. 2B. (It also contains an experiment asked by the other reviewer to compare to monoclonal anti-DNP Ab which indeed has not much of an effect)

5. A real concern is caused by Figure 2. In this figure the legend states that mice were vaccinated or not and injected with 4T1 tumor cells. When tumors were palpable day 7-8 the VEGF-DNP construct was injected. If this was the case why does the VEGF-DNP arm and OPN-DNP arm show reduced growth before the reagent is administered?

Our fault, problem was in the display. In panel A, the X-axis represented time from start of treatment with VEGF-DNP, which was day 7 from start of tumor implantation. Panel B is correct, X-axis represents days from tumor implantation. We corrected Fig. 2A in the revised version.

6. There seems to be no real attempt to address off-target toxicity. It seems very possible that there is likely to be some binding of the aptamers to non-tumor sources of VEGF and/or OPN, and also perhaps accumulation of the haptens in the liver. The manuscript would benefit from the authors addressing this concern. Ideally data could include imaging of organs following treatment with suitably labeled versions of the aptamers or hapten specific antibodies. If the authors feel this is beyond the scope of this manuscript then they should discuss toxicity concerns in their discussion.

In this proof-of-concept study we have not undertaken an in-depth analysis of potential toxicities, mostly because we have not seen overt toxicities like mortality, morbidity, weight loss or reduced movement in the cage. Nor have we seen toxicities in previous studies using the VEGF aptamer to target 4-1BB costimulation to tumors in which case we also looked for inflammatory infiltrated in various organs, in stark contrast to administration of a comparatively therapeutic dose of CTLA-4 (see reference 24 and Schrand et al., Can. Res., 2017, 77:1310). It would appear that any adverse effects, like for example formation and deposits of antibody-antigen complexes is minimal and subclinical. We have, therefore, discussed the toxicity issue briefly in the Discussion session (page 15, highlighted) stating:

“We have not observed overt toxicity in DNP vaccinated mice treated with VEGF-DNP in terms of mortality, morbidity, or weight loss. Nonetheless, we cannot exclude that subclinical levels of antibody-antigen aggregates have formed that may become clinically important upon intensification of therapy. “

7. Have the authors verified and validated the binding of their aptamer conjugates by flow cytometry on the various cell lines used? Please show data.

Yes. We stained with polyclonal anti-DNP antibody visualized with Alexa 647 labeled anti-murine IgG Ab. Binding though is quite weak, most likely because binding to the tumor cells (which do not express VEGF receptors) is mediated via weak interactions with heparan sulfate.

The rationale for VEGF targeting comes from our previous experience of targeting 4-1BB costimulation to tumors where we also show that VEGF conjugates accumulate in VEGF^{high} 4T1, but not VEGF^{low} 4T07, tumors (reference 24), and the studies showing that a proportion of tumor-expressed VEGF remain associated with tumor and adjacent ECM via binding to heparan sulfate (references 25-27). Consistent with that, VEGF-DNP or OPN-DNP were effective in the VEGF^{high} 4T1, but not VEGF^{low} 4T07 models (Extended Data Fig. 2).

8. Were Figure. 4A and B carried out side-by-side? In order to make this comparison they should be from the same experiment set up and presented on the same graph given that the FcR deficient mouse data are being compared to the wildtype with suitable statistical analysis.

The two experiments shown in panels A and B were done within the span of a week using the same reagents (we use aliquoted reagents kept frozen and used only once). Cannot present on the same graph because tumor growth in the FcR-deficient mice is slightly different, whether due to FcR defect or slight genetic drift not clear. Regardless, the main point is that the effect of therapy is lost in the FcR-deficient mice, that is we are comparing treatment to no treatment in wild type and FcR deficient mice.

9. Figure. 5. Do the authors have flow cytometry data demonstrating the depletion of the various lymphocyte populations? In the absence of these data it is difficult to ascertain the robustness of the lack of a role for CD8s.

Yes. We first titrate the purchased or in-house made antibodies, aliquot and freeze them, and use one or two aliquots per experiment once. Here is the depletion experiment using the antibody dose used in the study (experiment was done last year when the batches were received). It takes 1-2 weeks after the last administration before the lymphocyte counts begins to rise, and 4-5 weeks before it gets back to normal. (One also could argue that because antibody treatment abrogated tumor inhibition depletion must have been effective).

10. Figure 6 is difficult to interpret given the complexity of the system used; vaccination with a pig kidney homogenates which is a complex set of foreign antigens, a complex mixed mouse strain is used and an absence of the glycosyltransferase could introduce a wide range of both potential and missing antigens outside of aGal. These issues may well be beyond the scope of this paper but there should be some discussion of these issues and further there should be a little less emphasis placed on the concept of a lack of vaccination.

Notwithstanding the inevitable complexity of the experimental system as the reviewer pointed out, we can draw one conclusion – which was the primary purpose of the experiment: Tumor inhibition was dependent on (a) mice vaccinated against α Gal (and admittedly a slew of additional undefined antigens) and (b) administration of VEGF- α Gal. Taken together with the DNP studies, this is consistent with and supports the underlying hypothesis of this study that the observed inhibition was mediated by recruitment of α Gal antibodies to the tumor. Complicated and inadequate as it is, at present it is the only murine model available for the natural α Gal antibodies in humans. The tumor inhibition of day 3 B16BL6 tumors following monotherapy with VEGF- α Gal is rather remarkable (by historical comparisons) and to the best of our knowledge unprecedented. Tempting as it is, this is far from suggesting that in humans it will be also a potent therapy, largely because the complexities pointed out by the reviewer. Rather, that it may be worth to consider.

We agree that it is somewhat misleading to state that the above model recapitulates “lack of vaccination”. We have toned down this point changing the section title from “**Dispensing with vaccination by recruiting naturally occurring antibodies.**” to “**Recruiting anti- α Gal antibodies to tumors**” (Page 10, highlighted). Nowhere else do we make a reference to the suggestion that this model represents a model that dispenses with vaccination.

REVIEWERS' COMMENTS:

Reviewer #1 (Remarks to the Author):

The authors have satisfyingly addressed my questions.

Reviewer #2 (Remarks to the Author):

I am happy with the changes made to the manuscript and the thoughtful responses from the authors in these regard. I am satisfied that they have answered my comments and queries and feel that this manuscript is now acceptable for publication.